# Optimization of 2-Aminoquinazolin-4-(3*H*)-one Derivatives as Potent Inhibitors of SARS-CoV-2: Improved Synthesis and Pharmacokinetic Properties

**DOI:** 10.3390/ph15070831

**Published:** 2022-07-04

**Authors:** Young Sup Shin, Jun Young Lee, Sangeun Jeon, Jung-Eun Cho, Subeen Myung, Min Seong Jang, Seungtaek Kim, Jong Hwan Song, Hyoung Rae Kim, Hyeung-geun Park, Lak Shin Jeong, Chul Min Park

**Affiliations:** 1Research Institute of Pharmaceutical Sciences, College of Pharmacy, Seoul National University, Seoul 08826, Korea; pong1140@krict.re.kr (Y.S.S.); ljy3695@krict.re.kr (J.Y.L.); 2Center for Convergent Research of Emerging Virus Infection (CEVI), Korea Research Institute of Chemical Technology, 141 Gajeong-ro, Yuseong-gu, Daejeon 34114, Korea; jungeun@krict.re.kr (J.-E.C.); msb25@krict.re.kr (S.M.); jhsong@krict.re.kr (J.H.S.); hyungrk@krict.re.kr (H.R.K.); 3Zoonotic Virus Laboratory, Institut Pasteur Korea, Gyeonggi, Seongnam 13488, Korea; sangeun.jeon@ip-korea.org (S.J.); seungtaek.kim@ip-korea.org (S.K.); 4Medicinal Chemistry and Pharmacology, Korea University of Science and Technology, Daejeon 34114, Korea; 5Department of Non-Clinical Studies, Korea Institute of Toxicology, Yuseong-gu, Daejeon 34114, Korea; minseongjang@kitox.re.kr

**Keywords:** 2-aminoquinazolin-4-(3*H*)-one, SARS-CoV-2, COVID-19, pharmacokinetic properties, approximate lethal dose

## Abstract

We previously reported the potent antiviral effect of the 2-aminoquinazolin-4-(3*H*)-one **1,** which shows significant activity (IC_50_ = 0.23 μM) against severe acute respiratory syndrome coronavirus 2 (SARS-CoV-2) with no cytotoxicity. However, it is necessary to improve the in vivo pharmacokinetics of compound **1** because its area under the curve (AUC) and maximum plasma concentration are low. Here, we designed and synthesized *N*-substituted quinazolinone derivatives that had good pharmacokinetics and that retained their inhibitory activity against SARS-CoV-2. These compounds were conveniently prepared on a large scale through a one-pot reaction using Dimroth rearrangement as a key step. The synthesized compounds showed potent inhibitory activity, low binding to hERG channels, and good microsomal stability. In vivo pharmacokinetic studies showed that compound **2b** had the highest exposure (AUC_24h_ = 41.57 μg∙h/mL) of the synthesized compounds. An in vivo single-dose toxicity evaluation of compound **2b** at 250 and 500 mg/kg in rats resulted in no deaths and an approximate lethal dose greater than 500 mg/kg. This study shows that *N*-acetyl 2-aminoquinazolin-4-(3*H*)-one **2b** is a promising lead compound for developing anti-SARS-CoV-2 agents.

## 1. Introduction

Severe acute respiratory syndrome coronavirus 2 (SARS-CoV-2) has spread rapidly, caused the coronavirus disease 2019 (COVID-19) pandemic, and severely threatened global healthcare systems [1,2,3]. Since its first appearance in December 2019 in Wuhan, China, there have been over 533 million confirmed cases of COVID-19 worldwide as well as over 6.3 million deaths by 15 June 2022 [4].

SARS-CoV-2, which is an enveloped, positive-sense, single-stranded RNA beta-coronavirus, is one of seven coronaviruses known to infect humans. SARS-CoV, MERS-CoV, and SARS-CoV-2 cause severe acute respiratory tract infections, whereas HCoV-229E, HCoV-NL63, HCoV-OC43, and HCoV-HKU1 cause mild symptoms [5]. The genome sequence of SARS-CoV-2 is closely related to the sequence of SARS-like bat coronaviruses (bat-SL-CoVZC45 and bat-SL-CoVZXC21), sharing 88% identity, but has less genetic similarity to SARS-CoV (about 79%) and MERS-CoV (about 50%) [6].

Despite the effective COVID-19 vaccines that were developed at an unprecedented pace, SARS-CoV-2 variants, which spread more easily, continue to appear [7]. The transmission of the Delta variant, a lineage first detected in India in December 2020, is higher than that of the Alpha variant, which itself is more contagious than the original virus [8]. The Omicron variant, first discovered in South Africa in November 2021, is more infectious than the Delta variant and is more likely to lead to breakthrough infection in vaccinated individuals [9].

During the current COVID-19 pandemic, drug repurposing is the quickest way to find effective treatments, given that the safety profiles of repurposed drugs are generally well-established [10]. Several candidates have been investigated in large randomized clinical trials since early 2020. Although many drug candidates are being developed against COVID-19, there is currently no established treatment. Hydroxychloroquine and lopinavir–ritonavir, which received attention at the beginning of the pandemic, did not show good efficacy in clinical trials [11,12]. Remdesivir, which received emergency use approval from the US FDA, can be administered via IV to hospitalized patients and shortens recovery time but does not reduce mortality [13]. Molnupiravir, a prodrug of the nucleoside derivative *N*^4^-hydroxycytidine, is an orally available broad-spectrum antiviral agent that received emergency use approval by the US FDA in December 2021. However, results from phase 3 of the MOVe-OUT trial showed that the drug had modest effects. The primary outcome, the incidence of hospitalization or death at day 29, was 6.8% (48 out of 709 patients) in the molnupiravir group and 9.7% (68 out of 699 patients) in the placebo group, representing a 30% reduction in COVID-19-related hospitalization or death [14,15]. The potential for molnupiravir-induced mutagenesis remains a concern because its mechanism causes the accumulation of errors during RNA synthesis [16,17,18,19]. Additionally, in December 2021, the US FDA issued emergency use authorization for Paxlovid (two tablets of nirmatrelvir and one tablet of ritonavir). In the final analysis of the primary endpoint in the EPIC-HR phase 3 study, 0.7% of patients (5/697 hospitalized with no deaths) in the Paxlovid group were hospitalized, and 6.5% of patients (44/682 hospitalized with nine subsequent deaths) in the placebo group were hospitalized or died. Paxlovid reduced COVID-19-related hospitalization or death by 89%. [20]. It remains to be seen whether Paxlovid will be a game-changer given the potential for drug resistance or drug interactions. Therefore, in the absence of an agent with clear therapeutic benefits, new antiviral treatments for COVID-19 are still needed.

We recently reported that 2-aminoquinazolin-4-(3*H*)-one **1** has potent antiviral effects against SARS-CoV-2 and MERS-CoV [21,22]. Compound **1** showed significant activity (IC_50_ = 0.23 μM) against SARS-CoV-2 in a Vero cell assay, with no cytotoxicity and promising in vitro pharmacokinetic properties, including high microsomal stability and low binding to hERG channels.

However, it is necessary to improve the in vivo pharmacokinetics of compound **1** because its area under the curve (AUC) and maximum plasma concentration (C_max_) are low. These poor pharmacokinetic properties may be attributed to its planar structure. Therefore, our strategy in this study was to design *N*-substituted quinazolinone derivatives with good pharmacokinetics that retain their inhibitory activity against SARS-CoV-2 (Figure 1). Moreover, we attempted to conveniently synthesize aminoquinazolin-4-(3*H*)-one derivatives on a large scale. Here, we report the efficient synthesis of aminoquinazolin-4-(3*H*)-one derivatives and an assessment of the in vivo pharmacokinetics of an *N*-acetylated compound in rats.

## 2. Results and Discussion

### 2.1. Synthesis of 2-Aminoquinazolin-4-(3H)-one Derivatives

In the synthetic route used in our previous study, four steps were performed to synthesize key intermediate **1** with an overall yield of 50% [21,22]. The second step, which used an excess of POCl_3_, made the synthesis difficult because the product was unstable and easily returned to the starting material. For efficient and bulk synthesis, we used the one-pot reaction method of Tun-Cheng Chien’s group to synthesize intermediate **1** with an overall yield of 76% (Figure 1) [23]. Anthranilic acid **3** was treated with phenylcyanamide **4** and chlorotrimethylsilane to yield a mixture of 2-(*N*-substituted-amino)quinazolin-4-one **1** and 3-substituted 2-aminoquinazolin-4-one. A Dimroth rearrangement with 2 N sodium hydroxide in EtOH/H_2_O (1/1) afforded the key intermediate **1** [24]. This one-pot reaction, which was not sensitive to moisture, generated a large amount of the pure product **1** without the use of column chromatography. Demethylation of **1c** with boron tribromide afforded **5c**. Acetylation of **1a**, **1b**, and **5c** with acetic anhydride and triethylamine yielded *N*-acetylated compounds **2a**, **2b**, and **2c**, respectively. **1a** was treated with propionyl chloride, methyl chloroformate, iodomethane or p-toluenesulfonyl chloride, and triethylamine to yield the corresponding **6**. Compound **7** was also prepared by reacting compound **1a** and benzyloxyacetyl chloride with triethylamine. The hydrogenation of **7** in the presence of 10% Pd/C yielded alcohol **8**.

### 2.2. In Vitro Activity against SARS-CoV-2

The synthesized compounds were evaluated for antiviral activity against SARS-CoV-2. The nucleotide analog remdesivir was used as a reference compound (Table 1). The activity of the compounds was tested by immunofluorescence assay in Vero cells at 10 concentrations, starting at 25 μM in two-fold serial dilutions to determine the IC_50_ values [25]. Cytotoxicity (the CC_50_ value) was determined in parallel in uninfected cells. As shown in Table 1, acetylated derivatives **2a**, **2b**, and **2c** had potent inhibitory activity against SARS-CoV-2 (IC_50_ = 0.33, 0.29, and 0.11 μM, respectively). In addition, the acetylated compounds had lower ClogP values than compound **1a**, so it was expected that the pharmacokinetic properties would be improved. The propionyl **6a** and p-tosyl **6d** showed good activity (IC_50_ = 0.21 and 0.57 μM, respectively) but had ClogP values of 5.1 and 6.5, respectively, which lowered their priority for further investigation. The methyl carbamate **6b**, methyl **6c**, and hydroxyacetyl **8** showed only marginal antiviral activity (IC_50_ = 7.05, 5.66, and 2.57 μM, respectively). 

### 2.3. Mode of Action Study

The anti-SARS-CoV-2 activity of quinazolinones has been frequently reported upon, but the mode of action is still unidentified. Several promising targets for COVID-19 therapeutics have been reported, including angiotensin-converting Enzyme 2 (ACE-2), transmembrane protease serine subtype 2 (TMPRSS2), RNA-dependent RNA polymerase (RdRp), and 3-chymotrypsin-like protease (3CLpro) [3]. Among these targets, we focused on those involved in the viral entry mechanism.

To assess the effects of the compounds on viral entry, the SARS-CoV-2 spike pseudo-typed lentivirus system was used [26]. We tested the acetylated derivatives **2a** and **2b**, which showed good activity (IC_50_ = 0.54 and 0.99 μM, respectively) (Appendix A). The compounds inhibited the entry of the SARS-CoV-2 spike pseudo-typed virus in a concentration-dependent manner, which suggests that the compounds act as entry inhibitors.

### 2.4. Pharmacokinetics

Compounds **2a**–**c** were selected for in vivo pharmacokinetic studies based on their ClogP and selectivity index (SI) values. Each compound was administered orally at 10 mg/kg to three Sprague Dawley rats (Figure 2). The previously reported pharmacokinetic properties of compound **1a** in the rats showed a very low C_max_ value of 0.87 μg/mL and an AUC_24h_ of 6.62 μg∙h/mL. Compound **2a** had significantly higher exposure than compound **1,** with a C_max_ value of 1.85 μg/mL and an AUC_24h_ of 24.31 μg∙h/mL. Compound **2c** had an AUC_24h_ of only 10.77 μg∙h/mL, which was slightly better than that of compound **1a**. We anticipated that compound **2c** would have good absorption properties due to its low ClogP value; however, the experimental results showed that compound **2c** had lower exposure than compound **2b**. This result may be due to the poor solubility of compound **2c**. The pharmacokinetic results of compound **2b**, which was substituted with a difluoro moiety, showed that it had sufficient exposure, with a C_max_ value of 5.56 μg/mL and an AUC_24h_ of 41.57 μg∙h/mL and a blood concentration value that sufficiently exceeded the IC_50_ value within at least 8 h. The lipophilic properties and the small size of the fluorine moieties likely enhance the absorption of compound **2b**.

### 2.5. hERG Affinity, Microsomal Stability, and Cytotoxicity

Three acetylated compounds **2a-c** with strong activity against SARS-CoV-2 and good pharmacokinetic properties were assessed for hERG binding, microsomal stability, and cytotoxicity (Table 2). The results of the hERG patch–clamp assay showed that compounds **2a** and **2b** had weak activity against the hERG ion channel (IC_50_ = 15.2 and 30.0 μM, respectively). Compound **2c** was devoid of hERG activity (IC_50_ > 50 μM). Furthermore, the metabolic stability of each compound was determined by evaluating how much the compound remained in the liver microsomes after a 30 min activation of the metabolic enzyme system by NADPH. Compounds **2a** and **2b** had high stability in rat and human microsomes, whereas only 10% of compound **2c** was present in human microsomes at the end of the experiment. The hydroxyl group at position 5 of compound **2c** may contribute to this rapid metabolism [27]. Cytotoxicity was evaluated using an EZ-Cytox cell viability assay kit, which uses the water-soluble tetrazolium salt method [28]. Compound **2a** showed a CC_50_ value of less than 10 μM for all four normal cell lines (HFL-1, L929, NIH 3T3, and CHO-K1). Compounds **2b** and **2c** were relatively free from cytotoxicity. Overall, compound **2b** had the best safety and stability profiles in vitro.

### 2.6. In Vivo Single-Dose Toxicity

Given the potent activity of compound **2b** with no cytotoxicity and its promising pharmacokinetic profile, the acute toxicity of this compound was studied in male Sprague Dawley rats at two oral doses: 250 and 500 mg/kg (Figure 3). After single administration using 55% PEG200 in distilled water as a vehicle, each group of four rats was monitored for 15 days. The results of this single-dose toxicity study showed that there was no mortality in the treated or control rats. Observations of clinical signs showed that one animal in the 250 mg/kg group and two animals in the 500 mg/kg group had liquid feces but recovered by day 4. In addition, there was a statistically significant reduction (Dunnett’s test) in body weight on day 4 in the 250 mg/kg group and on days 4 and 8 in the 500 mg/kg group, but both groups recovered on day 14. Gross observations of the organs did not reveal changes in any of the organs examined. Therefore, the approximate lethal dose of compound **2b** in this study was estimated to be more than 500 mg/kg.

## 3. Materials and Methods

### 3.1. Chemistry

All of the reagents used in the experiments were purchased from commercial suppliers such as Aldrich, Tokyo Chemical Industry, Alfa aesar, or Combi-Blocks and used without further purification. Organic solvents were concentrated under reduced pressure using a Büchi rotary evaporator.

The progress of the reaction was confirmed by thin-layer chromatohgraphy (TLC) using plates coated with Kieselgel 60F_254_ (Merck), and column chromatography was carried out using RediSep Rf normal-phase silica flash columns (230–400 mesh).

Proton (^1^H) and carbon (^13^C) NMR spectra of the compounds were measured using a Bruker with each 300, 400, and 500 instrument at each 300, 400, and 500 MHz interval. Chemical shifts are reported as parts per million (δ) relative to the solvent peak. Resonance patterns are reported with the notations s (singlet), d (doublet), t (triplet), q (quartet), dd (doublet of doublets), and m (multiplet). Coupling constants (J) are reported in hertz. High-resolution mass spectra (HRMS) were obtained from the Korea Research Institute of Chemcial Technology using the FAB ionization method. Melting points were determined on a Metter Toledo MP50 instrument and are uncorrected.

#### 3.1.1. 7-Chloro-2-((3,5-dichlorophenyl)amino)quinazolin-4(3*H*)-one (**1a**)

Chlorotrimethylsilane (22 mL, 175 mmol) was added to a stirred suspension of **3a** (20 g, 117 mmol) and *N*-(3,5-dichlorophenyl)cyanamide **4a** (32.7 g, 175 mmol) in tert-butanol (300 mL) at rt. The reaction mixture was stirred using a mechanical stirrer for 4 h at 60 °C. To the stirred mixture, 2 N aqueous ethanolic NaOH (H_2_O/EtOH = 1/1, 1 L) was added at rt. The resulting solution was stirred for 6 h at 110 °C. The mixture was cooled to rt and acidified with the addition of acetic acid (114 mL) in an ice bath. The precipitated solid was collected and washed with H_2_O (200 mL), DCM (200 mL) and methanol (100 mL). The remaining solid was dried under a high vacuum to obtain **1a** (24.8 g, 62%) as a white solid: mp > 300 °C; ^1^H NMR (400 MHz, DMSO-d_6_) δ 11.24 (s, 1H, NH), 9.14 (s, 1H, NH), 7.97 (d, J = 8.5 Hz, 1H, arom.CH), 7.81 (s, 2H, arom.CH), 7.47 (d, J = 2.0 Hz, 1H, arom.CH), 7.30 (dd, J = 8.4, 2.1 Hz, 1H, arom.CH), 7.25 (t, J = 1.9 Hz, 1H, arom.CH) ppm; ^13^C NMR (100 MHz, DMSO-d_6_) δ 161.01 (C=O quinazolinone), 150.55 (C2 quinazolinone), 148.00 (C8a quinazolinone), 141.17 (C1 aniline), 139.18 (C7 quinazolinone), 134.05 (C5 quinazolinone), 127.89 (C3,5 aniline), 123.84 (C6 quinazolinone), 121.78 (C8 quinazolinone), 117.68 (C4 aniline), 114.68 (C4a quinazolinone), 113.33 (C2,6 aniline) ppm; HRMS (FAB) calculated for [C_14_H_8_Cl_3_N_3_O]^+^ ([M]^+^): 338.9733, observed: 339.9813 *m*/*z* [M + H]^+^.

#### 3.1.2. 7-Chloro-2-((3,5-difluorophenyl)amino)quinazolin-4(3*H*)-one (**1b**)

Following the same procedure used for the synthesis of **1a**, **3a** (4.8 g, 28 mmol), *N*-(3,5-difluorophenyl)cyanamide **4b** (6.5 g, 42 mmol) and chlorotrimethylsilane (5 mL, 42 mmol) were used to obtain **1b** (5 g, 58%) as a white solid: mp > 300 °C; ^1^H NMR (500 MHz, DMSO-d_6_) δ 11.08 (s, 1H, NH), 9.13 (s, 1H, NH), 7.94 (d, J = 8.4 Hz, 1H, arom.CH), 7.53–7.42 (m, 3H, arom.CH), 7.26 (dd, J = 8.4, 2.1 Hz, 1H, arom.CH), 6.85 (tt, J = 9.2, 2.4 Hz, 1H, arom.CH) ppm; ^13^C NMR (125 MHz, DMSO-d_6_) δ 163.47 (d, J = 15.7 Hz, C3 aniline), 161.54 (d, J = 15.5 Hz, C5 aniline), 160.98 (C=O quinazolinone), 150.61 (C2 quinazolinone), 147.95 (C8a quinazolinone), 141.33 (t, J = 14.0 Hz, C1 aniline), 139.19 (C7 quinazolinone), 127.84 (C5 quinazolinone), 124.67 (C6 quinazolinone), 123.83 (C8 quinazolinone), 117.45 (C4a quinazolinone), 102.28 (d, J = 29.1 Hz, C2,6 aniline), 97.61 (t, J = 26.2 Hz, C4 anline) ppm; HRMS (FAB) calculated for [C_14_H_8_ClF_2_N_3_O]^+^ ([M]^+^): 307.0324, observed: 307.0323 *m*/*z* [M]^+^.

#### 3.1.3. 2-((3,5-Dichlorophenyl)amino)-5-methoxyquinazolin-4(3*H*)-one (**1c**)

Following the same procedure used for the synthesis of **1a**, **3b** (20 g, 131 mmol), *N*-(3,5-dichlorophenyl)-cyanamide **4a** (36.6 g, 196 mmol) and chlorotrimethylsilane (25 mL, 196 mmol) were used to obtain **1c** (9.6 g, 24%) as a white solid: mp > 300 °C; ^1^H NMR (500 MHz, DMSO-d_6_) δ 10.74 (s, 1H, NH), 8.91 (s, 1H, NH), 7.81 (s, 2H, arom.CH), 7.55 (t, J = 8.2 Hz, 1H, arom.CH), 7.20 (s, 1H, arom.CH), 6.95 (d, J = 8.1 Hz, 1H, arom.CH), 6.79 (d, J = 8.3 Hz, 1H, arom.CH), 3.82 (s, 3H, OCH_3_) ppm; ^13^C NMR (125 MHz, DMSO-d_6_) δ 160.02 (C=O quinazolinone), 159.68 (C5 quinazolinone), 151.68 (C2 quinazolinone), 151.87 (C8a quinazolinone), 147.22 (C1 aniline), 141.56 (C7 quinazolinone), 134.88 (C3,5 anilne), 134.05 (C8 quinazolinone), 121.35 (C4 aniline), 117.47 (d, J = 50.1 Hz, C2,6 anilne), 108.33 (C6 quinazolinone), 105.76 (C4a quinazolinone), 55.81 (OCH_3_) ppm; HRMS (FAB) calculated for [C_14_H_9_Cl_2_N_3_O_2_]^+^ ([M]^+^): 321.0072, observed: 321.0058 *m*/*z* [M]^+^.

#### 3.1.4. *N*-(3,5-Dichlorophenyl)cyanamide (**4a**)

Cyanogen bromide (39 g, 370 mmol) and 1N NaOH (340 mL) were added to a stirred suspension of 3,5-dichloroaniline (50 g, 309 mmol) in acetic acid (150 mL) and water (1050 mL) at rt. The reaction mixture was stirred for 18 h at rt. The solid was filtered and washed with water (400 mL) and hexane (200 mL). The remaining solid was dried under a high vacuum to obtain **4a** (55.7 g, 97%) as a white solid: mp 182–184 °C; ^1^H NMR (400 MHz, DMSO-d_6_) δ 10.78 (s, 1H, NH), 7.27 (t, J = 1.8 Hz, 1H, arom.CH), 6.93 (d, J = 1.8 Hz, 2H, arom.CH) ppm; ^13^C NMR (100 MHz, DMSO-d_6_) δ 141.59 (C1 aniline), 135.16 (C3,5 aniline), 122.10 (CN), 113.77 (C4 aniline), 110.77 (C2,6 aniline) ppm; HRMS (FAB) calculated for [C_7_H_4_Cl_2_N_2_]^+^ ([M]^+^): 185.9752, observed: 185.9751 *m*/*z* [M]^+^.

#### 3.1.5. *N*-(3,5-Dichlorophenyl)cyanamide (**4b**)

Following the same procedure used for the synthesis of **4a**, 3,5-difluoroaniline (20 g, 155 mmol) and cyanogen bromide (20 g, 186 mmol) were used to obtain **4b** (23 g, 96%) as a white solid; mp 134–136 °C; ^1^H NMR (400 MHz, DMSO-d_6_) δ 6.91 (tt, J = 9.4, 2.2 Hz, 1H, arom.CH), 6.70–6.57 (m, 2H, arom.CH) ppm; ^13^C NMR (100 MHz, DMSO-d_6_) δ 164.20 (d, J = 15.4 Hz, C3 aniline), 162.24 (d, J = 15.5 Hz, C5 aniline), 141.97 (t, J = 13.4 Hz, C1 aniline), 110.88 (CN), 99.14–98.53 (m, C2,6 aniline), 98.01 (t, J = 26.1 Hz, C4 aniline) ppm; HRMS (FAB) calculated for [C_7_H_4_F_2_N_2_]^+^ ([M]^+^): 154.0343, observed: 154.0353 *m*/*z* [M]^+^.

#### 3.1.6. 2-((3,5-Dichlorophenyl)amino)-5-hydroxyquinazolin-4(3*H*)-one (**5c**)

To a stirred suspension of **1c** (4.5 g, 14.7 mmol) in anhydrous CH_2_Cl_2_ (150 mL) was added dropwise BBr_3_ (1.0 M solution in CH_2_Cl_2_, 44 mL, 44 mmol) at –78 °C. The reaction mixture was stirred for 3 h at rt. The mixture was quenched by the dropwise addition of methanol at 0 °C and concentrated under reduced pressure. The solid was filtered and washed with water, hexane, and methanol. The remaining solid was dried under a high vacuum to obtain **5c** (4 g, 97%) as a white solid; mp > 300 °C; ^1^H NMR (300 MHz, DMSO-d_6_) δ 11.55 (s, 2H, OH, NH), 9.25 (s, 1H, NH), 7.77 (s, 2H, arom.CH), 7.51 (t, J = 8.1 Hz, 1H, arom.CH), 7.26 (s, 1H, arom.CH), 6.83 (d, J = 8.1 Hz, 1H, arom.CH), 6.61 (d, J = 8.1 Hz, 1H, arom.CH) ppm; ^13^C NMR (125 MHz, DMSO-d_6_) δ 160.38 (C=O, C-OH quinazolinone), 141.45 (C2 quinazolinone), 136.54 (C8a quinazolinone, C1 aniline), 134.49 (C7 quinazolinone), 122.68 (C8 quinazolinone, C4 aniline), 118.87 (C6 quinazolinone), 109.91 (C2,6 aniline), 104.85 (C4a quinazolinone) ppm; HRMS (FAB) calculated for [C_14_H_9_Cl_2_N_3_O_2_]^+^ ([M]^+^): 321.0072, observed: 321.0058 *m*/*z* [M]^+^.

#### 3.1.7. *N*-(7-Dichloro-4-oxo-3,4-dihydroquinazolin-2-yl)-*N*-(3,5-dichlorophenyl)acetamide (**2a**)

To a stirred suspension of **1a** (5 g, 14.7 mmol) in CH_2_Cl_2_ (75 mL) were added trimethylamine (4.1 mL, 29.3 mmol) and acetic anhydride (4.2 mL, 44.0 mmol) at rt. The reaction mixture was stirred for 18 h at 40 °C. After the completion of the reaction (determined by TLC), the mixture was cooled to rt and diluted with CH_2_Cl_2_ (75 mL), then the mixture was washed with water (100 mL). The organic phase was concentrated under reduced pressure to obtain a crude solid, which was washed with CH_2_Cl_2_ (200 mL) and methanol (100 mL) to obtain **2a** (4.2 g, 75%) as a white solid; mp 236–238 °C; ^1^H NMR (500 MHz, DMSO-d_6_) δ 12.87 (s, 1H, NH), 8.09 (d, J = 8.4 Hz, 1H, arom.CH), 7.68 (t, J = 1.9 Hz, 1H, arom.CH), 7.62 (d, J = 2.1 Hz, 1H, arom.CH), 7.59 (d, J = 1.9 Hz, 2H, arom.CH), 7.54 (dd, J = 8.5, 2.1 Hz, 1H, arom.CH), 2.13 (s, 3H, CH_3_) ppm; ^13^C NMR (125 MHz, DMSO-d_6_) δ 171.39 (C=O acetyl), 161.92 (C=O quinazolinone), 149.54 (C2 quinazolinone), 149.07 (C8a quinazolinone), 141.62 (C1 aniline), 139.79 (C7 quinazolinone), 134.73 (C2,6 aniline), 128.46 (C5 quinazolinone), 128.45 (C3,5 aniline), 127.56 (C6 quinazolinone), 127.47 (C4 aniline), 126.77 (C8 quinazolinone), 120.18 (C4a quinazolinone), 24.29 (CH_3_ acetyl) ppm; HRMS (FAB) calculated for [C_16_H_10_Cl_3_N_3_O_2_]^+^ ([M]^+^): 380.9839, observed: 380.9838 *m*/*z* [M]^+^.

#### 3.1.8. *N*-(7-Chloro-4-oxo-3,4-dihydroquinazolin-2-yl)-*N*-(3,5-difluorophenyl)acetamide (**2b**)

To a stirred suspension of **1b** (151 mg, 0.49 mmol) in CH_2_Cl_2_ (2.5 mL) were added trimethylamine (137 μL, 0.98 mmol) and acetic anhydride (139 μL, 1.47 mmol) at rt. The reaction mixture was stirred for 5 h at 40 °C. The mixture was cooled to rt and diluted with CH_2_Cl_2_ (20 mL), then the mixture was washed with water (20 mL). The organic phase was dried over MgSO_4_, filtered, and concentrated under reduced pressure to obtain the crude compound, which was purified by silica gel column chromatography (Hx/EA 5/1) to produce **2b** (124 mg, 72%) as a white solid; mp 178–180 °C; ^1^H NMR (700 MHz, acetone-d_6_) δ 12.45 (s, 1H, NH), 8.08 (d, J = 8.5 Hz, 1H, arom.CH), 7.40 (dd, J = 8.5, 2.0 Hz, 1H, arom.CH), 7.36 (dd, J = 7.3, 2.3 Hz, 2H, arom.CH), 7.29 (d, J = 2.1 Hz, 1H, arom.CH), 7.26 (ddd, J = 9.2, 6.8, 2.4 Hz, 1H, arom.CH), 2.18 (s, 3H, CH_3_) ppm; ^13^C NMR (175 MHz, acetone-d_6_) δ 175.96 (C=O acetyl), 165.45 (d, J = 14.6 Hz, C3 aniline), 164.04 (d, J = 14.2 Hz, C5 aniline), 161.61 (C=O quinazolinone), 151.28 (C3 quinazolinone), 151.12 (C8a quinazolinone), 143.20 (t, J = 12.9 Hz, C1 aniline), 141.56 (C7 quinazolinone), 129.59 (C5 quinazolinone), 127.72 (C6 quinazolinone), 127.53 (C8 quinazolinone), 120.60 (C4a quinazolinone), 115.12 (dd, J = 22.2, 5.3 Hz, C2,6 anline), 106.24 (t, J = 25.8 Hz, C4 aniline) 26.84 (CH_3_ acetyl) ppm; HRMS (FAB) calculated for [C_16_H_10_ClF_2_N_3_O_2_]^+^ ([M]^+^): 349.0430, observed: 349.0426 *m*/*z* [M]^+^.

#### 3.1.9. *N*-(3,5-Dichlorophenyl)-*N*-(5-hydroxy-4-oxo-3,4-dihydroquinazolin-2-yl)acetamide (**2c**)

To a stirred suspension of **5c** (100 mg, 0.31 mmol) in CH_2_Cl_2_ (2 mL) were added trimethylamine (86 μL, 0.62 mmol) and acetic anhydride (87 μL, 0.93 mmol) at rt. The reaction mixture was stirred for 18 h at 40 °C. The mixture was cooled to rt and diluted with CH_2_Cl_2_ (20 mL), then the mixture was washed with water (20 mL). The organic phase was dried over MgSO_4_ and filtered and concentrated under reduced pressure to obtain a crude compound, which was purified by silica gel column chromatography (Hx/EA 4/1) to produce **2c** (64 mg, 57%) as a white solid; mp > 300 °C; ^1^H NMR (500 MHz, DMSO-d_6_) δ 13.09 (s, 1H, OH), 11.59 (s, 1H, NH), 7.68 (t, J = 1.9 Hz, 1H, arom.CH), 7.64 (t, J = 8.2 Hz, 1H, arom.CH), 7.60 (s, 2H, arom.CH), 7.03 (d, J = 8.1 Hz, 1H, arom.CH), 6.86 (d, J = 8.2 Hz, 1H, arom.CH), 2.13 (s, 3H, COCH_3_) ppm; ^13^C NMR (125 MHz, DMSO-d_6_) δ 170.72 (C=O acetyl), 166.94 (C=O quinazolinone), 159.69 (C-OH quinazolinone), 148.49 (C2 quinazolinone), 146.77 (C8a quinazolinone), 141.25 (C1 aniline), 136.38 (C7 quinazolinone), 134.31 (C2,6 aniline), 127.96 (C3,5 aniline), 126.89 (C4 aniline), 117.02 (C8 quinazolinone), 112.07 (C6 quinazolinone), 107.07 (C4a quinazolinone), 23.65 (CH_3_ acetyl) ppm; HRMS (FAB) calculated for [C_15_H_8_Cl_3_N_3_O_2_]^+^ ([M]^+^): 366.9682, observed: 366.9684 *m*/*z* [M]^+^.

#### 3.1.10. *N*-(7-Chloro-4-oxo-3,4-dihydroquinazolin-2-yl)-*N*-(3,5-dichlorophenyl)propionamide (**6a**)

Following the same procedure used for the synthesis of **2b**, **1a** (100 mg, 0.29 mmol), trimethylamine (82 μL, 0.59 mmol) and propionyl chloride (55 μL, 0.88 mmol) were used to obtain **6a** (61 mg, 52%) as a white solid: mp 223–225 °C; ^1^H NMR (500 MHz, DMSO-d_6_) δ 12.87 (s, 1H, NH), 8.09 (d, J = 8.5 Hz, 1H, arom.CH), 7.70 (t, J = 1.9 Hz, 1H, arom.CH), 7.62 (d, J = 2.0 Hz, 1H, arom.CH), 7.59 (d, J = 1.9 Hz, 2H, arom.CH), 7.55 (dd, J = 8.5, 2.1 Hz, 1H, arom.CH), 2.37 (q, J = 7.3 Hz, 2H, CH_2_), 1.03 (t, J = 7.3 Hz, 3H, CH_3_) ppm; ^13^C NMR (125 MHz, DMSO-d_6_) δ 174.42 (C=O propionyl), 161.49 (C=O quinazolinone), 149.16 (C2 quinazolinone), 148.65 (C8a quinazolinone), 140.91 (C1 aniline), 139.38 (C7 quinazolinone), 134.31 (C2,6 aniline), 128.08 (C5 quinazolinone), 128.06 (C3,5 aniline), 127.22 (C6 quinazolinone), 127.10 (C4 aniline), 126.34 (C8 quinazolinone), 119.69 (C4a quinazolinone), 28.75 (CH_2_ propionyl), 8.87 (CH_3_ propionyl) ppm; HRMS (FAB) calculated for [C_17_H_12_Cl_3_N_3_O_2_]^+^ ([M]^+^): 394.9995, observed: 395.0020 *m*/*z* [M]^+^.

#### 3.1.11. Methyl (7-Chloro-4-oxo-3,4-dihydroquinazolin-2-yl)(3,5-dichlorophenyl)carbamate (**6b**)

To a stirred suspension of **1a** (200 mg, 0.59 mmol) in THF (6 mL) were added trimethylamine (123 μL, 0.88 mmol) and 4-dimethylaminopyridine (7.2 mg, 0.059 mmol) at rt. After stirring for 30 min, methyl chloroformate (68 μL, 0.88 mmol) was added dropwise to the reaction mixture at 0 °C. The resulting solution was stirred for 18 h at rt. After completion of the reaction (measured by TLC), the solution was diluted with ethyl acetate (40 mL), and the mixture was washed with water (20 mL). The organic phase was dried over MgSO_4_ and filtered and concentrated under reduced pressure to obtain a crude compound, which was purified by silica gel column chromatography (Hx/EA 5/1) to produce **6b** (49 mg, 21%) as a white solid; mp 218–220 °C; ^1^H NMR (400 MHz, DMSO-d_6_) δ 12.81 (s, 1H, NH), 8.10 (d, J = 8.5 Hz, 1H, arom.CH), 7.60 (t, J = 1.9 Hz, 1H, arom.CH), 7.59 (d, J = 2.1 Hz, 1H, arom.CH), 7.57 (d, J = 1.9 Hz, 2H, arom.CH), 7.53 (dd, J = 8.5, 2.1 Hz, 1H, arom.CH), 3.77 (s, 3H, OCH_3_) ppm; ^13^C NMR (100 MHz, DMSO-d_6_) δ 161.47 (C=O quinazolinone), 153.20 (C2 quinazolinone), 149.10 (C=O methylester), 148.17 (C8a quinazolinone), 141.01 (C7 quinazolinone), 139.34 (C1 aniline), 134.00 (C2,6 aniline), 128.01 (C5 quinazolinone), 127.12 (C3,5 aniline), 127.02 (C6 quinazolinone), 126.14 (C4 aniline), 125.91 (C8 quinazolinone), 119.58 (C4a quinazolinone), 54.30 (CH_3_ methylester) ppm; HRMS (FAB) calculated for [C_16_H_10_Cl_3_N_3_O_3_]^+^ ([M]^+^): 396.9788, observed: 396.9777 *m*/*z* [M]^+^.

#### 3.1.12. 7-Chloro-2-((3,5-dichlorophenyl)(methyl)amino)quinazolin-4(3*H*)-one (**6c**)

To a stirred suspension of **1a** (100 mg, 0.3 mmol) in DMF (2.5 mL) was added sodium hydride (60% dispersion mineral oil, 18 mg, 0.45 mmol) at rt. After stirring for 30 min, iodomethane (22 μL, 0.36 mmol) was added dropwise to the reaction mixture at 0 °C. The resulting mixture was stirred for 2 h at rt. After the completion of the reaction (determined by TLC), the solution was diluted with ethyl acetate (20 mL), and the mixture was washed with water (3 × 10 mL). The organic phase was dried over MgSO_4_ and filtered and concentrated under reduced pressure to obtain the crude compound, which was purified by silica gel column chromatography (Hx/EA 4/1) to produce **6c** (49 mg, 47%) as a white solid; mp 287–289 °C; ^1^H NMR (500 MHz, DMSO-d_6_) δ 11.52 (s, 1H, NH), 7.92 (d, J = 8.4 Hz, 1H, arom.CH), 7.63–7.44 (m, 3H, arom.CH), 7.36 (d, J = 2.1 Hz, 1H, arom.CH), 7.22 (dd, J = 8.4, 2.1 Hz, 1H, arom.CH), 3.41 (s, 3H, CH_3_) ppm; ^13^C NMR (125 MHz, DMSO-d_6_) δ 162.54 (C=O quinazolinone), 151.28 (C8a quinazolinone), 146.09 (C1 aniline), 138.91 (C2 quinazolinone), 134.53 (C7 quinazolinone), 128.04 (C5 quinazolinone), 126.02 (C3,5 aniline), 125.51 (C2,6 aniline), 123.93 (C8 quinazolinone), 123.19 (C4a quinazolinone), 116.92 (C4 aniline) ppm; HRMS (FAB) calculated for [C_15_H_10_Cl_3_N_3_O]+ ([M]^+^): 352.9889, observed: 352.9879 *m*/*z* [M]^+^.

#### 3.1.13. *N*-(7-Chloro-4-oxo-3,4-dihydroquinazolin-2-yl)-*N*-(3,5-dichlorophenyl)-4-methylben-zenesulfonamide (**6d**)

To a stirred suspension of **1a** (100 mg, 0.29 mmol) in CH_2_Cl_2_ (3 mL) was added trimethylamine (82 μL, 0.59 mmol) at rt. After stirring for 30 min, p-toluenesulfonyl chloride (168 μL, 0.88 mmol) was added dropwise to the reaction mixture at 0 °C. The resulting solution was stirred for 3 h at 40 °C. After the completion of the reaction (determined by TLC), the mixture was cooled to rt and diluted with CH_2_Cl_2_ (20 mL), and then the mixture was washed with water (20 mL). The organic phase was dried over MgSO_4_ and filtered and concentrated under reduced pressure to obtain a crude compound, which was purified by silica gel column chromatography (Hx/EA 9/1) to produce **6d** (116 mg, 80%) as a white solid; mp 163–165 °C; ^1^H NMR (500 MHz, DMSO-d_6_) δ 7.96 (d, J = 8.5 Hz, 1H, arom.CH), 7.78 (d, J = 1.9 Hz, 2H, arom.CH), 7.53–7.50 (m, 2H, arom.CH), 7.46 (d, J = 2.0 Hz, 1H, arom.CH), 7.31 (dd, J = 8.5, 2.0 Hz, 1H, arom.CH), 7.27 (t, J = 1.9 Hz, 1H, arom.CH), 7.16–7.13 (m, 2H, arom.CH), 2.29 (s, 3H, CH_3_) ppm; ^13^C NMR (125 MHz, DMSO-d_6_) δ 160.99 (C=O quinazolinone), 148.31 (C2 quinazolinone), 144.84 (C8a quinazolinone), 140.78 (C1 aniline), 139.37 (C7 quinazolinone), 138.30 (C4 tosyl), 134.19 (C1 tosyl), 128.35 (C5 quinazolinone), 128.10 (C3,5 aniline), 125.56 (C2,6 tosyl), 124.18 (C6 quinazolinone), 123.45 (C8 quinazolinone), 122.51 (C4a quinazolinone), 118.55 (C2,6 aniline), 117.21 (C4 aniline), 20.87 (CH_3_ tosyl) ppm; HRMS (FAB) calculated for [C_21_H_14_Cl_3_N_3_O_3_S]^+^ ([M]^+^): 492.9821, observed: 492.9840 *m*/*z* [M]^+^.

#### 3.1.14. 2-(Benzyloxy)-*N*-(7-chloro-4-oxo-3,4-dihydroquinazolin-2-yl)-*N*-(3,5-dichlorophenyl)-2-hydroxy-acetamide (**7**)

To a stirred suspension of **1a** (3 g, 8.8 mmol) in CH_2_Cl_2_ (50 mL) was added trimethylamine (2.5 mL, 17.6 mmol) at rt. After stirring for 30 min, benzyloxyacetyl chloride (4.4 mL, 26.4 mmol) was added dropwise to the reaction mixture at 0 °C. The resulting solution was stirred for 3 h at 40 °C. After the completion of the reaction (determined by TLC), the mixture was washed and cooled to rt and diluted with CH_2_Cl_2_ (100 mL). The mixture was then washed with water (100 mL). The organic phase was dried over MgSO_4_ and filtered and concentrated under reduced pressure to obtain a crude compound, which was purified by silica gel column chromatography (Hx/EA 5/1) to produce **7** (4.2 g, 98%) as a colorless oil; mp 282–284 °C; ^1^H NMR (300 MHz, acetone-d_6_) δ 12.15 (s, 1H, NH), 8.11 (d, J = 8.5 Hz, 1H, arom.CH), 7.65 (d, J = 1.3 Hz, 3H, arom.CH), 7.46 (dd, J = 8.5, 2.1 Hz, 1H, arom.CH), 7.40 (d, J = 2.0 Hz, 1H, arom.CH), 7.33–7.25 (m, 5H, arom.CH), 4.58 (s, 2H, CH_2_), 4.32 (s, 2H, CH_2_) ppm; ^13^C NMR (125 MHz, acetone-d_6_) δ 173.77 (C=O acetamide), 161.17 (C=O quinazolinone), 150.19 (C2 quinazolinone), 149.82 (C8a quinazolinone), 140.87 (C7 quinazolinone), 140.11 (C1 benzyloxy), 138.56 (C2,6 aniline), 135.91 (C5 quinazolinone), 129.85 (C6 quinazolinone), 129.08 (d, J = 3.7 Hz, C3,5 benzyloxy), 128.71 (t, J = 23.6 Hz, C4 aniline), 127.13 (d, J = 10.0 Hz, C8 quinazolinone), 120.20 (C4a quinazolinone), 73.73 (CH_2_ benzyloxy), 70.59 (CH_2_ acetamide) ppm; HRMS (FAB) calculated for [C_23_H_16_Cl_3_N_3_O_3_]^+^ ([M]^+^): 487.0257, observed: 487.0260 *m*/*z* [M]^+^.

#### 3.1.15. *N*-(7-Chloro-4-oxo-3,4-dihydroquinazolin-2-yl)-*N*-(3,5-dichlorophenyl)-2-hydroxy-acetamide (**8**)

To a stirred solution of **7** (3 g, 6.1 mmol) in ethyl acetate (70 mL) was added 10% palladium on activated carbon (1.2 g) at rt. The reaction mixture was stirred under H2 for 2 h at rt. After the completion of the reaction (determined by TLC), the mixture was diluted with CH_2_Cl_2_ (200 mL) and methanol (200 mL), and the suspension was then filtered through a celite pad and concentrated to produce alcohol **8** (194 mg, 8%) as a white solid; mp 263–265 °C; ^1^H NMR (500 MHz, DMSO-d_6_) δ 12.73 (s, 1H, OH), 10.59 (s, 1H, NH), 8.01 (d, J = 8.5 Hz, 1H, arom.CH), 7.66 (d, J = 1.9 Hz, 2H, arom.CH), 7.43 (d, J = 2.1 Hz, 1H, arom.CH), 7.38 (dd, J = 8.5, 2.0 Hz, 1H, arom.CH), 7.31 (t, J = 2.0 Hz, 1H, arom.CH), 5.05 (s, 2H, CH_2_) ppm; ^13^C NMR (125 MHz, DMSO-d_6_) δ 166.19 (C=O hydroxy acetamide), 161.95 (C=O quinazolinone), 154.21 (C2 quinazolinone), 149.28 (C8a quinazolinone), 140.77 (C1 aniline), 139.30 (C7 quinazolinone), 134.26 (C2,6 aniline), 134.08 (C5 quinazolinone), 128.26 (C6 quinazolinone), 124.87 (d, J = 20.9 Hz, C3,5 aniline), 122.94 (C4 aniline), 118.22 (C8 quinazolinone), 117.43 (C4a quinazolinone), 64.91 (CH_2_ hydroxyacetamide) ppm; HRMS (FAB) calculated for [C_16_H_10_Cl_3_N_3_O_3_]^+^ ([M]^+^): 396.9788, observed: 396.9767 *m*/*z* [M]^+^.

### 3.2. Cell Line and Virus

Vero cells (ATCC CCL-81, Manassas, VA, USA) were maintained at 37 °C with 5% CO_2_ in Dulbecco’s Modified Eagle’s medium (DMEM; Welgene, Gyeongsan-si, Korea) supplemented with 10% heat-inactivated fetal bovine serum (FBS) and 1X antibiotic-antimycotic solution (Gibco/Thermo Fisher Scientific, Waltham, MA, USA). SARS-CoV-2 (βCoV/KOR/KCDC03/2020) was supplied by the Korea Centers for Disease Control and Prevention (KCDC). This virus was propagated in Vero cells, and viral titers were determined by plaque assays in the Vero cells. All experiments related to SARS-CoV-2 were conducted at laboratories of the Institut Pasteur Korea in accordance with the guidelines of the Korea National Institute of Health (KNIH) using enhanced Biosafety Level 3 (BSL-3) containment procedures approved for use by the KCDC [25].

### 3.3. Concentration–Response Curve Analysis by Immunofluorescence Assay

Vero cells were seeded at 1.2 × 10^4^ cells in 384-well, μClear plates (Greiner Bio-One, Kremsmünster, Austria) 24 h prior to the experiment. Compounds were placed in intermediate 384-well plates containing serum-free media (Gibco, Waltham, MA, USA). The diluted compounds (10 concentrations, 0.05–25 μM) were added to the cell plates in 10 μL volumes (at a final DMSO concentration of 0.5% (*v*/*v*)). For viral infection, plates were moved into the BSL-3 containment facility, and SARS-CoV-2 was added to the Vero cells at a multiplicity of infection (MOI) of 0.0125. At 24 h post-infection, the cells were fixed with 4% paraformaldehyde and stained with SARS-CoV-2 nucleocapsid protein, and immunofluorescence analysis was performed. To quantify the number of cells and infection ratios, Columbus software (PerkinElmer, Waltham, MA, USA) was used, and antiviral activity was normalized to the positive (mock-infected) and negative (0.5% DMSO) controls in each assay plate. Concentration–response curves were fitted by sigmoidal models using XLfit 4 in Microsoft Excel or GraphPad Prism 6.0 software (GraphPad Software, San Diego, CA, USA). The IC_50_ values were calculated from curves fitted to the normalized activity dataset. The quality of each assay was controlled by the Z′-factor [25].

### 3.4. Pseudovirus-Based Entry Assay

For the SARS-CoV-2 pseudovirus entry assay, H1299 (human lung squamous carcinoma cell) cells stably expressing ACE2 and TMPRSS2 were seeded in white, 384-well, μClear plates (Greiner Bio-One) so that they reached 70% confluency the following day. The cells were pre-treated with serially diluted compounds and mixed with SARS-CoV-2 pseudovirus particles. After incubation for 24 h, the culture medium was replaced with fresh DMEM containing 2% FBS. Pseudovirus entry efficiency was quantified by measuring luciferase activity in infected cells 48 h after infection using a Bright-Glo luciferase assay system (Promega, Madison, WI, USA). The relative luminescence units were normalized to the positive (0.5% DMSO, set as 100%) and negative (mock-infected, set as 0%) controls, and the IC_50_ values were calculated using GraphPad Prism 6.0 software. Cell viability was measured using a CellTiter-Glo luminescent cell viability assay (Promega, Madison, WI, USA) according to the manufacturer’s instructions [26].

### 3.5. hERG Channel Patch-Clamp Assay

hERG-HEK293 cells (Genionics, Zürich, Switzerland) were cultured in DMEM/F-12 medium (10% FBS, 7.5 mg/mL, blasticidin S HCl, 50 mg/mL hygromycin B). Cells (2 × 10^5^) were seeded in a 60 mm cell-culture dish. After culturing for 2 days in an incubator at 5% CO_2_ and 37 °C, the prepared cells and Sealchips (Aviva Biosciences, San Diego, CA, USA) were loaded into a PatchXpress 7000A (Molecular Devices, San Jose, CA, USA) patch-clamp system, and a protocol for measuring hERG channel activity was conducted. A pressure of −35 to −47 mmHg was applied to position the cells in the micro-holes of the Sealchip. A negative pressure of 0 to −45 mmHg was repeatedly applied until a seal resistance of 1 GΩ was obtained, and a negative pressure of −130 mmHg was applied to form a whole-cell patch. A MultiClamp 700A microelectrode amplifier was used for voltage-clamping and data acquisition, and DataXpress software 2.0.4.5 (Molecular Devices) was used to measure the tail current amplitude to determine hERG channel activity. Data were analyzed using Clampfit 9.2 software (Molecular Devices).

### 3.6. Microsomal Stability Assay

The phase I stability of the compounds was measured by quantifying (using LC/MS/MS) the amount of compound remaining after exposure to liver microsomes in which the metabolic enzyme system was activated by NADPH for 30 min. The activity of the enzyme system was confirmed using buspirone as a reference compound (<10% of buspirone remained in human microsomes, <5% of buspirone remained in rat microsomes) [29].

### 3.7. Cell Viability Assay

An EZ-Cytox assay (DoGenBio, Seoul, Korea) was used to determine cell viability. Cells (2 × 10^4^ cells) were seeded in a 96-well plate and cultured in RPM1640 supplemented with 10% FBS in a 5% CO_2_ incubator at 37 °C. After 24 h, 100, 10, 1, 0.1, and 0.01 μM of each compound was added to the wells. After 24 h, 10 μL of the WST reagent solution was added to each well. After incubation for 3 h, absorbance was measured at 450 nm using a microplate reader. Cell viability was determined as the percentage of the measured value in the presence of compound compared with the DMSO control, and the CC_50_ value was obtained from the cell viability plot in GraphPad prism 6 [28].

## 4. Conclusions

To improve the in vivo pharmacokinetics of 2-aminoquinazolin-4-(3*H*)-one compounds, we synthesized a series of *N*-substituted derivatives. These compounds could be conveniently prepared on a large scale through a one-pot reaction using Dimroth rearrangement as a key step. All of the *N*-acetylated compounds showed good potency and favorable properties such as low hERG binding and good microsomal stability. The results of the pharmacokinetic studies in rats showed that compound **2b** had the highest AUC_24h_ (41.57 μg∙h/mL), and the blood concentration sufficiently exceeded the IC_50_ value within at least 8 h. Furthermore, the in vivo toxicity of compound **2b** was assessed over 15 days following a single dose of either 250 or 500 mg/kg. The results showed that no animals died and that the approximate lethal dose is likely more than 500 mg/kg. Consequently, this study shows that compound **2b** is a very promising lead compound for developing anti-SARS-CoV-2 agents.

## Data Availability

The data presented in this study are available in the article.

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
