# Peer review of "Optimization of 2-Aminoquinazolin-4-(3H)-one Derivatives as Potent Inhibitors of SARS-CoV-2: Improved Synthesis and Pharmacokinetic Properties"

_pharmaceuticals, 2022, doi:10.3390/ph15070831_

Round 1

Reviewer 1 Report

The manuscript entitled" Optimization of 2-Aminoquinazolin-4-(3H)-one Derivatives as Potent Inhibitors of SARS-CoV-2: An Efficient Synthesis and Improved Pharmacokinetic Properties" seems to be a good research article. However, please consider the following points:

1- The English language of the manuscript should be revised.

2- The introduction should be improved

3- Certain methods should be more clarified in the manuscript like antiviral assay. 

Author Response

We are very thankful to the reviewer for the careful review and constructive comments.

Point 1: The English language of the manuscript should be revised.

Response 1: We made efforts to improve manuscript further, and the language was revised with the help of MDPI's English pre-edit service and Bioedit's English proofreading service.

Point 2: The introduction should be improved.

Response 2: In response to the reviewer’s comment, we have rewritten the introduction section in the manuscript to describe more aptly content that fits the topic by referring to recent papers and eliminating outdated content.

[Page 1]

SARS-CoV-2, which is an enveloped, positive-sense, single-stranded RNA beta-coronavirus, is one of seven coronaviruses that are known to infect humans. SARS-CoV, MERS-CoV, and SARS-CoV-2 cause severe acute respiratory tract infections, whereas HCoV-229E, HCoV-NL63, HCoV-OC43, and HCoV-HKU1 cause mild symptoms [5].

[Page 2]

During the current COVID-19 pandemic, drug repurposing is the quickest way to find effective treatments, given that the safety profiles of repurposed drugs are generally well-established [10].

Point 3: Certain methods should be more clarified in the manuscript like antiviral assay.

Response 3: Considering the comment by the reviewer, Sections '3.2. Cell line and virus' and '3.3. Concentration-response curve analysis by immunofluorescence assay' were added to describe the antiviral assay in more detail.

[Page 13]

3.2. Cell line and virus

Vero cells (ATCC CCL-81, Manassas, VA, USA) were maintained at 37 °C with 5% CO2 in Dulbecco’s Modified Eagle’s medium (DMEM; Welgene, Gyeongsan-si, Korea), supplemented with 10% heat-inactivated fetal bovine serum (FBS) and 1X antibiotic-antimycotic solution (Gibco/Thermo Fisher Scientific, Waltham, MA, USA). SARS-CoV-2 (βCoV/KOR/KCDC03/2020) was supplied by the Korea Centers for Disease Control and Prevention (KCDC). This virus was propagated in Vero cells and viral titers were determined by plaque assays in Vero cells. All experiments related to SARS-CoV-2 were conducted at laboratories of the Institut Pasteur Korea in accordance with the guidelines of the Korea National Institute of Health(KNIH) using enhanced Biosafety Level 3 (BSL-3) containment procedures approved for use by the KCDC [22].

3.3. Concentration–response curve analysis by immunofluorescence assay

Vero cells were seeded at 1.2 × 104 cells in 384-well, μClear plates (Greiner Bio-One, Kremsmünster, Austria) 24 h prior to the experiment. Compounds were placed in intermediate 384-well plates containing serum-free media (Gibco, Waltham, MA, USA). The diluted compounds (10 concentrations, 0.05–25 μM) were added to the cell plates in 10 L volumes (at a final DMSO concentration of 0.5% (v/v)). For viral infection, plates were moved into the BSL-3 containment facility, and SARS-CoV-2 was added at a multiplicity of infection(MOI) of 0.0125 to the Vero cells. At 24 h post-infection, the cells were fixed with 4% paraformaldehyde and stained with SARS-CoV-2 nucleocapsid protein, and immunofluorescence analysis was performed. To quantify the number of cells and infection ratios, Columbus software (PerkinElmer, Waltham, MA, USA) was used, and antiviral activity was normalized to positive (mock-infected) and negative (0.5% DMSO) controls in each assay plate. Concentration–response curves were fitted by sigmoidal models using XLfit 4 in Microsoft Excel or GraphPad Prism 6.0 software (GraphPad Software, San Diego, CA, USA). The IC50 values were calculated from curves fitted to the normalized activity dataset. The quality of each assay was controlled by the Z′-factor [22].

Reviewer 2 Report

The authors submitted a manuscript focused on the study of aminoquinazoline derivatives against SARS-CoV-2. In particular, the work starts from previous results of the group on an active aminoquinazoline compound which was modified to improve its properties. The authors studied chemical aspects, pharmacokinetic and pharmacodynamic properties of the compounds, but the paper does not meet the standard for publication in Pharmaceuticals at this stage. I report some comments below.

·      List of authors: check last name! The list finishes with “and…”.

·  The title is also puzzling: “An Efficient Synthesis and Improved Pharmacokinetic Properties” does not sound correct. Should it be “improved synthesis and pk properties”? Or authors should refer to the optimization of the reference compound…

·      Intoduction: the first part (until line 49) sounds outdated. Authors must take into account that several contributions appeared in the literature focused on this topic and many are more updated.

·      Figure 1 and its caption are somehow misleading: the compound did not only undergo N-derivatization, but Cl was also replaced by F.

·      Scheme 1 describes the synthesis quite clearly, but it is not cited within the text.

·      Compound charachterization: in NMR transcripts, signal attribution should be added.

·      Quality of figures (2 and 3) should be improved: now they just appear as plain outputs of MS Excel with standard colors.

·      The author tested hERG affinity, microsomal stability, and cytotoxicity of the compounds. Nevertheless, there is no discussion on the paper about how the potential mechanism of action of these compounds was identified.

·      The manuscript must be checked for typos (e.g. comma at the end of title 2.4)

·      Overall, paper formatting is not in line with the template and font is different among parts.

Author Response

Response: We appreciate the reviewer’s thoughtful comments to our manuscript. We have revised the manuscript according to the reviewer’s comments.

Point 1: List of authors: check last name! The list finishes with “and…”.

Response 1: We thank the reviewer for the correct comment. All typos related to author names have been corrected.

[Page 1]

Citation: Shin, Y.S.; Lee, J.Y.; Jeon, S.; Cho, J.-E.; Myung, S.; Jang, M.S.; Kim, S.; Song, J.H.; Kim, H.R.; Park, H.-g.; Jeong, L.S.; Park, C.M.

Young Sup Shin1,2,, Jun Young Lee1,2,, Sangeun Jeon3, Jung-Eun Cho2, Subeen Myung2, Min Seong Jang4, Seungtaek Kim3, Jong Hwan Song2, Hyoung Rae Kim2, Hyeung-geun Park1,*, Lak Shin Jeong1,*, and Chul Min Park2,*

Point 2: The title is also puzzling: “An Efficient Synthesis and Improved Pharmacokinetic Properties” does not sound correct. Should it be “improved synthesis and pk properties”? Or authors should refer to the optimization of the reference compound…

Response 2: As the reviewer suggested, we have rewritten the title so that it clearly conveys the concepts explained in the paper.

[Page 1]

Optimization of 2-Aminoquinazolin-4-(3H)-one Derivatives as Potent Inhibitors of SARS-CoV-2: Improved Synthesis and Pharmacokinetic Properties

Point 3: Introduction: the first part (until line 49) sounds outdated. Authors must take into account that several contributions appeared in the literature focused on this topic and many are more updated.

Response 3: We agree with the reviewer's comment that the first part of the introduction is outdated, and these contents have been reinforced by more recent papers and data.

[Page 1]

SARS-CoV-2, which is an enveloped, positive-sense, single-stranded RNA beta-coronavirus, is one of seven coronaviruses that are known to infect humans. SARS-CoV, MERS-CoV, and SARS-CoV-2 cause severe acute respiratory tract infections, whereas HCoV-229E, HCoV-NL63, HCoV-OC43, and HCoV-HKU1 cause mild symptoms [5].

[Page 2]

During the current COVID-19 pandemic, drug repurposing is the quickest way to find effective treatments, given that the safety profiles of repurposed drugs are generally well-established [10].

Point 4: Figure 1 and its caption are somehow misleading: the compound did not only undergo N-derivatization, but Cl was also replaced by F.

Response 4: As the reviewer mentioned, given that compound 1a was not only N-derivatized, we have redrawn the figure to avoid any misunderstanding. The novel structures synthesized by derivatization are shown in detail.

[Page 3]

Figure 1. Design of novel 2-aminoquinazolinone compounds to improve in vivo pharmacokinetics.

Point 5: Scheme 1 describes the synthesis quite clearly, but it is not cited within the text.

Response 5: We thank the reviewer for the correct comment. Scheme 1 is now referred to within the text.

[Page 3]

…For efficient and bulk synthesis, we used the one-pot reaction method of Tun-Cheng Chien’s group to synthesize intermediate 1 with an overall yield of 76% (Scheme 1) [20]. Anthranilic acid 3 was treated with phenylcyanamide 4 and chlorotrimethylsilane to yield a mixture…

Point 6: Compound characterization: in NMR transcripts, signal attribution should be added.

Response 6: As the reviewer pointed out, there were errors in some of the NMR transcripts. All modifications have been completed and each signal attribution is now added. As a representative example, the modified NMR script of 1c is described below.

[Page 9]

1H NMR (500 MHz, DMSO-d6) δ 10.74 (s, 1H, NH), 8.91 (s, 1H, NH), 7.81 (s, 2H, arom.CH), 7.55 (t, J = 8.2 Hz, 1H, arom.CH), 7.20 (s, 1H, arom.CH), 6.95 (d, J = 8.1 Hz, 1H, arom.CH), 6.79 (d, J = 8.3 Hz, 1H, arom.CH), 3.82 (s, 3H, OCH3) ppm; 13C NMR (125 MHz, DMSO-d6) δ 160.02 (C=O quinazolinone), 159.68 (C5 quinazolinone), 151.68 (C2 quinazolinone), 151.87 (C8a quinazolinone), 147.22 (C1 aniline), 141.56 (C7 quinazolinone), 134.88 (C3,5 anilne), 134.05 (C8 quinazolinone), 121.35 (C4 aniline), 117.47 (d, J = 50.1 Hz, C2,6 anilne), 108.33 (C6 quinazolinone), 105.76 (C4a quinazolinone), 55.81 (OCH3) ppm;

Point 7: Quality of figures (2 and 3) should be improved: now they just appear as plain outputs of MS Excel with standard colors.

Response 7: We thank the reviewer for the helpful comment. Figures 2 and 3 have been redrawn using OriginPro8.5.

Point 8: The author tested hERG affinity, microsomal stability, and cytotoxicity of the compounds. Nevertheless, there is no discussion on the paper about how the potential mechanism of action of these compounds was identified.

Response 8: We have reported several effective quinazoline compounds against SARS-CoV-2, but the mechanism was not elucidated. Considering this constructive criticism, we therefore attempted to find targets of quinazolinone compounds, focusing on the viral entry part among previously known targets. In addition, by performing a cell-based pseudovirus entry assay, we found that our compounds act as viral entry inhibitors.

[Page 4]

To assess the effects of the compounds on viral entry, the SARS-CoV-2 spike pseudotyped lentivirus system was used [23]. We tested the acetylated derivatives 2a and 2b, which showed good activity (IC50 = 0.54 and 0.99 μM, respectively) (Figure S1). The compounds inhibited the entry of the SARS-CoV-2 spike pseudotyped virus in a concentration-dependent manner, which suggests that the compounds act as entry inhibitors.

The additional data were added in the revised supporting information:

Figure 1S. Concentration-response inhibition curves of 2a and 2b against SARS-CoV-2 pseudovirus

Point 9. The manuscript must be checked for typos (e.g. comma at the end of title 2.4)

Point 10. Overall, paper formatting is not in line with the template and font is different among parts.

 Response 9 and 10: We thank the reviewer for the correct comment. Comma at the end of title 2.4 has been removed. The template and font have been modified to fit the format of Pharmaceuticals.

[Page 6]

2.5. hERG affinity, microsomal stability, and cytotoxicity

Round 2

Reviewer 2 Report

The authors modified the manuscript as suggested.